# Intelligent, In-Vehicle Autonomous Decision-Making Functionality for Driving Style Reconfigurations

**Ilias Panagiotopoulos * and George Dimitrakopoulos**

Informatics and Telematics Department, Harokopio University of Athens, 17778 Athens, Greece
* Correspondence: ipanagio@hua.gr

**Abstract:** Intelligent connected vehicles (ICVs) constitute a transformative technology attracting immense research effort and holding great promise in providing road safety, transport efficiency, driving comfort, and eco-friendly mobility. As the driving environment becomes more and more "connected", the manner in which an ICV is driven (driving style) can dynamically vary from time to time, due to the change in several parameters associated with personal traits and with the ICV's surroundings. This necessitates fast and effective decisions to be made for a priori identifying the most appropriate driving style for an ICV. Accordingly, the main goal of this study is to present a novel, in-vehicle autonomous decision-making functionality, which enables ICVs to dynamically, transparently, and securely utilize the best available driving style (DS). The proposed functionality takes as input several parameters related to the driver's personal characteristics and preferences, as well as the changing driving environment. A Naive Bayes learning classifier is applied for the cognitive nature of the presented functionality. Three scenarios, with regards to drivers with different personal preferences and to driving scenes with changing environment situations, are illustrated, showcasing the effectiveness of the proposed functionality.

**Keywords:** cognitive decision making; context-aware; driving style; intelligent connected vehicles; Naive Bayes classifier; self-adaptive functionality





## 1. Introduction

Intelligent connected vehicles (ICVs) continue to attract immense research interest [1–4], as they constitute a transformative technology that holds a great promise in providing road safety, transport efficiency, driving comfort, and eco-friendly mobility.

Recent advances in ICV research are reflected on driver assistance systems, which include functionalities such as navigation assistance, adaptive cruise control, traffic detection, emergency braking, important vehicle notifications, driver alerts, parking assistance, etc. [5,6]. In addition, due to the fact that several parameters related to a vehicle's vicinity can dynamically change in an unforeseen manner, ICVs should have the ability to adapt quickly, making effective decisions [7,8]. In addressing these challenges, cognitive management principles and machine learning techniques have provided the opportunity to model in-vehicle autonomous decision-making functionalities, by aggregating various data from all the types of sensors in order to enhance knowledge building [9,10]. Some of the most commonly machine learning techniques that have been applied include Bayesian networks (BNs), artificial neural networks (ANNs), and support vector machines (SVMs) [11,12].

A specific factor that affects vehicular behavior is the, so called, driving style (DS), which describes a set of parameters crucial for the way a vehicle behaves, such as the suspension system, the gearbox change frequency, etc. [13]. ICVs dispose multidimensional driving styles, based on data from the in-vehicle sensors, the communication related data from other vehicles or/and road infrastructure units, as well as on data relevant to the driver's personal preferences. Moreover, as the driving environment becomes more and more "connected", the DS can dynamically change, due to the change in several parameters

associated with personal traits and with the ICV's surroundings. This necessitates fast and effective decisions to be made for a priori identifying the most appropriate driving style for an ICV.

Therefore, an in-vehicle autonomous decision-making functionality that can provide the best available driving style (DS), is gradually becoming a significant challenge that can help ICVs increase their driving safety and riding comfort factors. Such functionality must be able to combine different input parameters with respect to the complex external environment and the driver's preferences. All these attributes/parameters can be changing, in a random manner, and can significantly affect the DS selection. Additionally, driving the ICVs can affect the importance of these attributes/parameters, which in turn influences the communication between the driving environment and the ICVs [14,15].

This paper accordingly builds on several research efforts and describes a novel functionality for the dynamic reconfiguration of driving styles, namely 'i-DSS' (intelligent-driving style selection), incorporating knowledge and experience. In this respect, its contribution lies exactly in the following elements:

- It presents a novel in-vehicle decision-making functionality, able to proactively, efficiently, and securely decide on the most appropriate DS, in dynamically changing environments, considering all of the driver's personal preferences, as well as contextual parameters from the vehicle's environment.
- The proposed functionality acts in a fully autonomous (self-adaptive) manner, requiring no driver intervention.
- It follows a human-centric approach, where cognitive management techniques are incorporated to aggregate extensive data sources in real-time (driving surrounding context, driver's preferences, and operational requirements) and interprets them to assess whether a specific DS is appropriate in each case.
- It provides decisions with increased reliability, through applying Bayesian networking concepts, and specifically the Naïve Bayes (NB) classifier, which attributes a cognitive nature to the functionality [16–18].

The structure of the paper is as follows. Section 2 includes a comprehensive literature review. Section 3 describes the problem description and formulation, whereas Section 4 presents the overall methodology approach through the exploitation of Bayesian networks theory, and specifically the multiclass NB classifier. Section 5 showcases the effectiveness of 'i-DSS', implementing a set of distinct case studies, followed by comprehensive simulation results. Concluding remarks and future research plans are depicted in Section 6.

## 2. Literature Review and Background Work

This section presents the background work on ICVs and relevant research areas, identifying some gaps that pave the way and motivate the present work.

### 2.1. Research Areas and Achievements in ICVs

Decision making has a pivotal role in ICVs, as it bridges perception with control, reconfiguring various vehicular operating parameters and helping drivers to cope with hazardous conditions [19]. The performance of ICVs reflected on transport efficiency, driving comfort, road safety, and energy consumption is highly affected by the in-vehicle decision-making process [20]. Sensing technologies have an indispensable part in constructing efficient in-vehicle computing systems and autonomous decision-making functionalities, which in turn aim to enhance the successful deployment of ICVs in the near future [21,22].

The real time DS adaptation to the changing environment conditions and driver intentions/preferences involves high levels of novel research. Following these research efforts, it can be stated that collecting as much data as possible from the vehicle's surrounding movement and the driver's state, real-time personalized DS decisions can improve the driver's interaction with ICVs [23,24].

On this basis, ICVs should be programmed to behave in a trustful and acceptable manner, i.e., considering that most of the people tend to trust agents that behave in a similar

manner to humans [25]. On this basis, drivers should be able to inform their ICVs how they want to be driven in order to enhance their driving experience inside them. This statement is based on the fact that "driving enjoyment" is one of the main determinants affecting consumers' future desire to use and accept ICVs [26,27]. Additionally, ICV's operator system needs a high amount of information from the external changing environment [28–30].

Following the above, the study of Ma and Zhang [31] investigated the effects of the driver's DS and designed DS of an automated vehicle on driver's acceptance, trust, and takeover behavior. The major implication of this study shows that driver's acceptance and trust towards automated vehicles is enhanced when the in-vehicle designed DS is aligned with driver's DS, and thus, undesired takeover behaviors are reduced. Feng et al. [32] proposed a novel data-driven fuzzy logic-based approach, where different driving styles are simulated, by using specified environmental inputs associated with the human driver perception. Extensive evaluation results in a unified comparative environment showcase the differences in DS, as well as their influence on fuel consumption. Moreover, Chen et al. [33] proposed a Bayesian model by incorporating the prior knowledge to understand DS recognitions. The results provide information about the driving behaviors for each DS, and therefore, this technique can be applied in individualization connected and automated driving.

In addition, the work of Sun et al. [34] presents a new approach for identifying user driving behaviors in an automated vehicle, and therefore, adapts its DS accordingly through a comprehensive software–hardware co-analysis. Furthermore, Hang et al. [35] present a decision-making framework, within the context of autonomous driving, where different states of DS and social interaction parameters are taken into consideration with respect to travel comfort, road efficiency, and vehicle safety. Two game theory approaches were adopted for addressing the decision-making problem, i.e., the Stackelberg and the Nash equilibrium games. Testing scenarios focusing on lane change maneuvers showed the efficiency of the developed analysis in making reasonable choices under different driving styles and providing personalized decisions for different drivers. Furthermore, Galante et al. [36] investigated road users' behavior, in the presence of low radius curves, via simulator experiments, whereas Eboli et al. [37] explored the relationships among DS and drivers' characteristics such as somatic, behavioral and emotional conditions. Brijs et al. [38] presents a new advanced driver assistance system (ADAS) which supports drivers as they overtake cyclists in order to avoid or, at least, mitigate crashes. This is followed by a multistage warning system via multiple modalities. Last, the social acceptance of autonomous vehicles is a matter of investigation (see e.g., [39]) with still many open topics to deal with.

### 2.2. Bayesian Networks

Computationally intensive real-time decision-making operations are undesirable in ICVs. To deal with this requirement, Bayesian networking concepts, seem as a valid candidate to be applied for the cognitive nature of the presented functionality. Bayesian networks (BNs) theory is a well-established supervised machine learning technique for modelling real-life problems can be transformed to simpler problems, by focusing on specific attributes of interest [40]. On this basis, BNs can be used effectively in classification decision-making processes, within the frame of ICVs, in uncertain external environments with various levels of complexity. On this basis, the in-vehicle operator system can act in a better way than human drivers, being able to adapt to the requirements of complex driving scenes, as well as to driver's profile and status characteristics.

In particular, in this paper the Naïve Bayes (NB) classifiers are used [16–18], aiming to increase the reliability of its decisions. The implementation of the NB-supervised machine learning classifier aims to maintain certain levels of simplicity, without affecting the extendibility and effectiveness of the proposed analysis. Furthermore, testing scenarios are used by making reasonable choices under different driving styles and providing personalized decisions for drivers with different preferences and motivations and external driving scenes with different environment characteristics.

## 3. Problem Description and Formulation

This section aims at presenting the problem description with respect to the DS adaptations for ICVs, which can dynamically vary from time to time, due to the change in several parameters associated with personal traits and with the external driving environment. Additionally, this section sets up and formulates a representative business case, being able to reflect the potential applications of the proposed 'i-DSS' functionality.

### 3.1. Problem Description

In general, the manner in which an ICV operates, the external driving environment, and the driver's profile and status, can be changed over time. On this basis, changes in the DS are very likely to need to occur while the ICV is moving.

As already mentioned, 'i-DSS' is capable of proposing each time the most appropriate reconfiguration of the DS, with respect to a set of predefined criteria, based on a set of requirements, as follows:

1.  Personalization: this requirement ensures that the proposed decisions are adapted to the driver's needs;
2.  Adaptability: this requirement supports the efficient interaction with the users, with respect to their personality characteristics and personal preferences;
3.  Knowledge aggregation: this requirement aims to accelerate future decisions based on the information extracted from past interactions;
4.  Scalability: this allows the appropriate adaptation of the decisions, based on the particular contextual needs.

The proposed 'i-DSS' consists of several functional modules, the rationale of which is described in detail below, first with regards to the input module and then with regards to the solution module. The above are shown on Figure 1, which illustrates 'i-DSS at a high level. More in detail, the input module combines the following three information sources:

1.  Quality of service (QoS): these parameters are associated with the performance of the proposed 'i-DSS' functionality, such as comfort, economy, vehicle control, vehicle reaction, etc.;
2.  Profile and status of the driver: these parameters are associated with a specific driver of the ICV and their personality characteristics, such as driving experience, gender, age, gender, mental state, etc.;
3.  External driving environment: these parameters are associated with real-time crucial information obtained from infrastructure units or/and other ICVs, such as road type, vehicle congestion level, road condition, etc. Most of these data are impossible or difficult to measure directly from the in-vehicle sensors.

In the present study, the input module of 'i-DSS' functionality uses two sets of policies, with respect to the importance level of the above input information sources. Policies are based on two main sets. The first set refers to driver's motivations towards a set of predetermined QoS parameters, as well as towards a set of predefined driver's personal characteristics. A driver can use a scale between 0 and 1 to provide importance values to each one of the above parameters. The lowest importance refers to value 0, whereas the highest importance refers to value 1. On the other hand, the second set refers to the cognitive nature and design restrictions of the in-vehicle central operator system towards a number of predefined driving environment features. Like previously, the ICV's operator system puts numerical weight values to each one of the predefined driving environment features. As these levels of importance may change over time during the ICV's ride, its operator system needs to adapt the corresponding numerical values frequently.

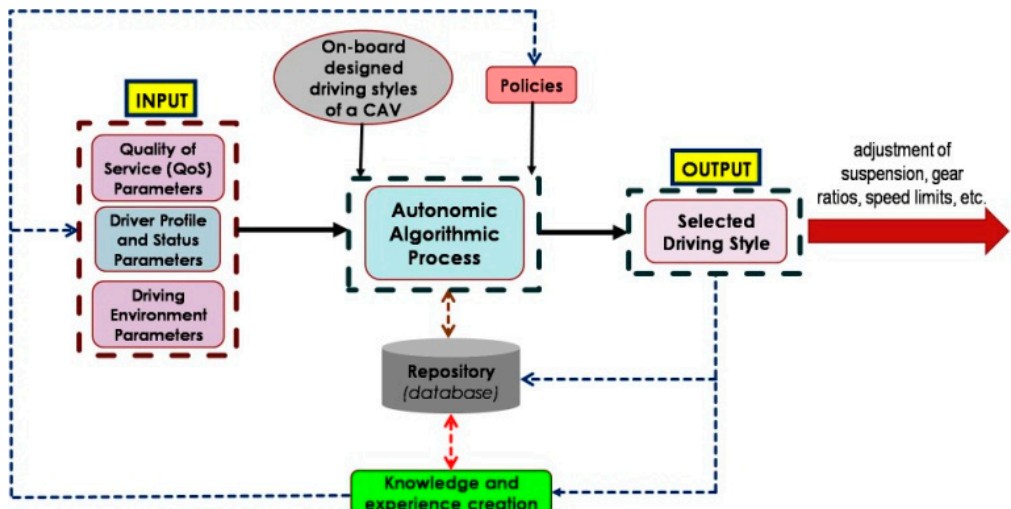

**Figure 1.** 'i-DSS' functional architecture.

Moreover, the input scheme of 'i-DSS' includes a knowledge building module, which aims to support the decision-making process. This module includes two large-scale data sets representing dependencies between DS selections and input information sources. More in detail, the first data set refers to an evaluation process, where drivers rate the behavior characteristics of the 'i-DSS', after their rides with ICVs in past. Essentially it represents dependencies between DS selections and input QoS parameters. Furthermore, the second data set refers to dependencies among DS selections and input external environment parameters through the in-vehicle operator system of ICVs. These dependencies are based on previous runs and decisions of the 'i-DSS' functionality.

One of the most important modules of the 'i-DSS' is the solution module, which is responsible for the selection and reconfiguration of DS. The solution module is designed to operate through the application of Bayesian networking principles, and specifically, through the exploitation of the NB-supervised machine learning classifier, as presented in detail in section III. More in detail, the embedded autonomic algorithmic process uses all the available information from the input module (including driver's profile and status, QoS parameters, external environment parameters, policies, and experience-based scheme) and optimizes an objective function (OF) towards the optimal DS. As such, the output scheme of the 'i-DSS' includes all the necessary actions that should be made for the reconfiguration of the DS, i.e., change in suspension, adjustment of gear ratios, alteration of speed vehicle, etc.

All DS reconfiguration decisions, alongside contextual information used as an input, are stored in a well-structured repository (database). Every time a contextual situation is encountered, the system compares it with information available in the database. In the case of a match above a certain percentage (e.g., 95%), the system decides on re-applying a previously successful decision, without running again the time-consuming algorithmic process described above. In a negative case, the algorithm needs to run whatsoever.

*3.2. Business Case*

As extracted from the architecture above, it is very important to set up a representative business case, being able to reflect the potential applications of 'i-DSS'. A driver wishes to have a ride with their ICV (owned or shared). Furthermore, a predefined set of designed driving styles is included in the operator system of the ICV. The driver should have an authorized account to access 'i-DSS', either through the central console of the vehicle or through their mobile phone. At the first time, where the driver logs on 'i-DSS', a well-established form should be filled in (in the form of numerical weight values) about driver's personality characteristics (e.g., driving experience, age, gender, etc.), as well as driver's preferences towards a predefined set of QoS features. Of course, the driver can change all these values, at any time, and whenever he/she wishes.

All the above data are stored in the repository (database) for each specific driver, as many individual drivers with different personality characteristics can use the same ICV for their travels when needed (e.g., family, car-sharing services, car-pooling services, etc.). In such case, 'i-DSS' has the ability to recognize the driver and access their past activity, preferences, and profile characteristics.

Following the above description, three use cases have been identified to showcase the efficiency of the proposed 'i-DSS' and its decision analysis, in dynamically providing the optimal DS during the ICV's ride. Drivers with different preferences and external environments with different features have been taken into consideration for the execution of the extensive simulation analysis.

## 4. Methodology

This section aims to present the proposed framework and the context in which the optimization solution approach (selection and reconfiguration of DS) is designed to operate, through the exploitation of Bayesian networking concepts and specifically the NB-supervised machine learning classifier. The algorithmic process uses the contextual input information and optimizes an objective function (OF) towards the optimal DS to be chosen.

### 4.1. Proposed Framework and Parameters Selection

In general, BNs express probabilistic relations among a set of predefined variables of interest. In Figure 2, such a BN is illustrated for the modeling of the decision-making process in the 'i-DSS' functionality. The variable DS is the child node (represents predictor selection) and refers to the driving style and its available states. Based on the study of Wang and Lukic [41], DS is classified into the following three distinct states: aggressive (DS = 1), normal (DS = 2), and defensive (DS = 3).

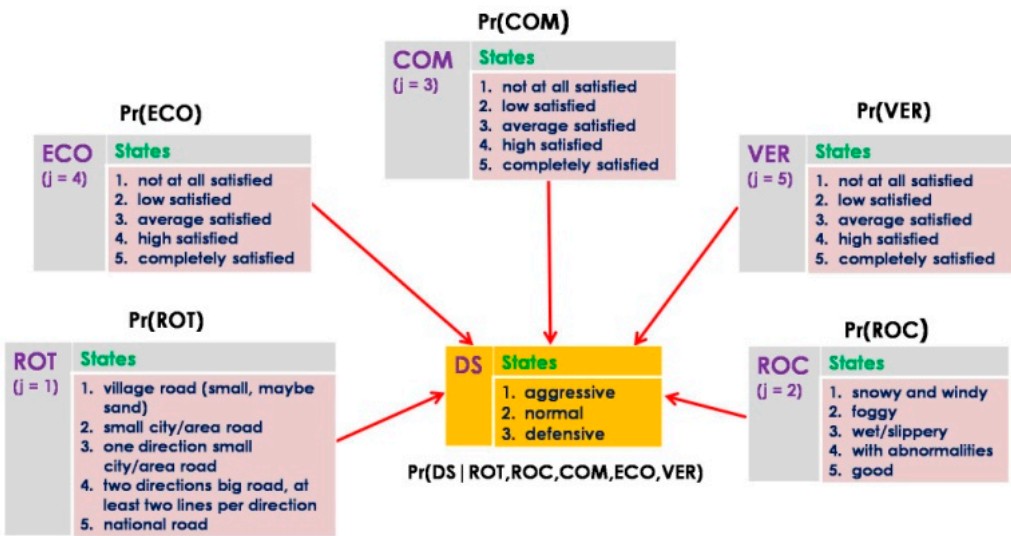

**Figure 2.** Bayesian network for DS prediction in ICVs.

Aggressive DS is the state consisting of harsh acceleration and deceleration abrupt speed change, and risky speeding [24], whereas the defensive DS refers to careful driving and smooth acceleration and deceleration [25].

As the present analysis aims to minimize the complexity of the solution approach in 'i-DSS', five (M = 5) parent nodes are considered, as depicted in Figure 2:

- two parameters associated with the driving environment scene; road condition (ROC) and road type (ROT);
- three QoS parameters; vehicle reaction (VER), comfort (COM), and economy (ECO)



It should be stated that the proposed 'i-DSS' is highly scalable, that means that can be easily adapted to various input parameters. In the present analysis, features which are related to the driver's profile and status have not been taken into consideration.

### 4.2. Knowledge-Based Process

An important factor in the DS selection has to do with the reconfiguration process and learning building, which helps 'i-DSS' to obtain knowledge and experience gradually. This process is based on the calculation of the conditional probabilities, which have the form $Pr[V_j = rs_{ij}{}^k \mid DS = i]$, where $rs_{ij}{}^k$ represents the k-th reference state for the *j*-th parameter ($j = 1, \ldots, $ M), when the *i*-th state ($i = 1, 2, 3$) is considered for the variable DS.

In the present study, discrete reference states are considered for each parameter *j*, as shown in Figure 2. In addition, according to the NB classifier, the input variables ROC, ROT, ECO, COM and VER are supposed to be statistically independent.

Following the above, the class-conditional posterior probability can be written as follows:

$$Pr[Y = k/x_1, \ldots, x_n] = \frac{Pr[Y = k] \bullet Pr[x_1, \ldots, x_n/Y = k]}{Pr[x_1, \ldots, x_n]} = \frac{f(x_i)}{Pr[x_1, \ldots, x_n]} \quad (1)$$

given the class *k* of the variable *Y* and the n-dimensional observation vector $X = (x_1, x_2, \ldots, x_n)$. The numerator in the right-hand side of (1) refers to the probability density function $f(x_i)$, where the knowledge about the contextual input scheme is considered. On this basis, the term $f(x_i)$ expresses the probability to achieve a certain contextual input scheme, when the class 'k' is considered for the variable *Y*.

Following the above, the present learning process updates the term $f(x_i)$, which can be written as follows:

$$f(x_i) = Pr[DS = i].\prod_{j=1}^{M} Pr[V_j = rs_{ij}^k \mid DS = i] \quad (2)$$

In general, the higher the value of the $f(x_i)$, the more reliable the existing information is for the specific value of *DS*, and therefore, the reliability of *DS* selections is enhanced.

According to (2), $Pr[DS = i]$ represents the prior probabilities, which is related to the volume of data that already exists for each state *i* of the *DS*. It should be stated that the sum of all the $Pr[DS = i]$ is equal to 1. The calculation of $Pr[DS = i]$ is based on the data collected for all the available states *i* of the *DS*, through the large-scale data sets described previously in Section 2. In other words, the calculation of each $Pr[DS = i]$ is based on the following formula:

$$Pr[DS = i] = \frac{count[DS = i]}{count[DS = 1] + count[DS = 2] + count[DS = 3]} \quad (3)$$

Additionally, the conditional probabilities $Pr[V_j = rs_{ij}{}^k \mid DS = i]$ in (2) are constantly being updated through the following formula:

$$Pr[V_j = rs_{ij}^k \mid DS = i]_{new} = L_{ij} \cdot cf_{ij}^k \cdot Pr[V_j = rs_{ij}^k \mid DS = i]_{old} \quad (4)$$

where the correction factor $cf_{ij}{}^k$ and the normalization factor $L_{ij}$ play important role in the adaptation of the conditional probabilities [33].

### 4.3. Selection Scheme

The selection scheme refers to the selection of the most appropriate DS state, among the available in-vehicle DS states, based on the probabilities of achieving the input parameter states. Figure 3 shows an overview of the selection process, which uses the input contextual level and finds the optimal DS with respect to the optimization of a well-defined objective function (OF).

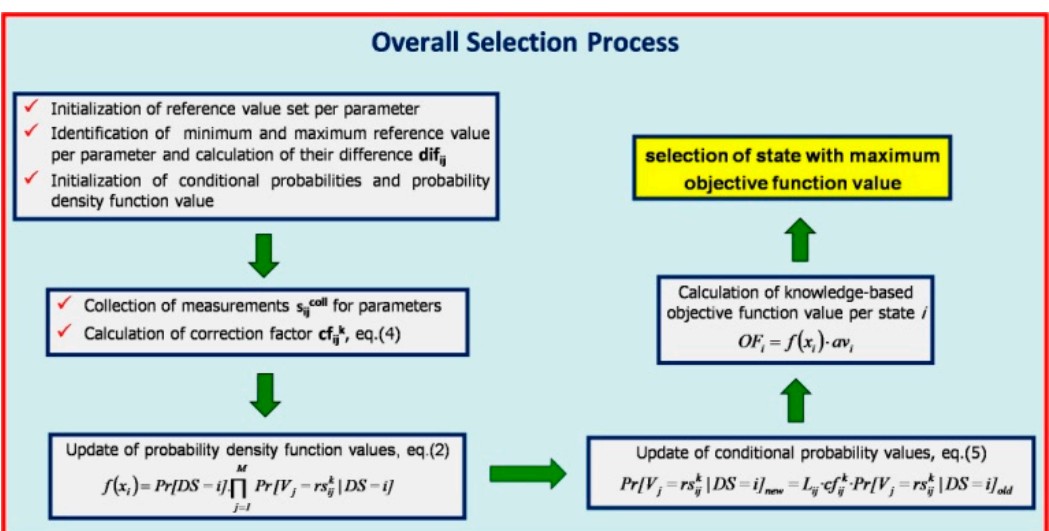

**Figure 3.** Overview of 'i-DSS' selection process.

In the present analysis, a new factor is introduced for each state $i$ of the variable DS, the appropriateness value $av_i$. Its computation is based on the highest, $hs_j$, and lowest, $ls_j$, states for each input parameter $j$ ($j = 1, \dots, M$). Additionally, the $rs_{ij}^k$ values are normalized through the following formula:

$$nrs_{ij}^k = \frac{\left|rs_{ij}^k - ls_j\right|}{\left|hs_j - ls_j\right|} \tag{5}$$

The normalized $nrs_{ij}^k$ can be equal to one when the lowest, $ls_j$, and highest, $hs_j$, values are equal. Taking into consideration the importance level $w_j$ of each input parameter $j$, the appropriateness value $av_i$ for a specific state $i$ can be computed:

$$av_i = \sum_{j=1}^{M} w_j nrs_{ij}^k \tag{6}$$

As mentioned previously, the selection scheme is based on the calculation of an objective function for each state $i$ of the $DS$, namely $OF_i$, based on the appropriateness value $av_i$ and the probability density function value $f(x_i)$:

$$OF_i = f(x_i) \cdot av_i \tag{7}$$

According to (7), all the available DS are classified and the one state with the highest $OF_i$ value is selected. Finally, the ICV moves on with the selected DS, through the in-vehicle operator system, by making all the necessary actions, i.e., alteration of vehicle speed, adjustment of gear ratios, etc. The above analysis enhances the efficiency of the overall solution approach in generating quick DS outputs, by reducing the computational time and the amount of memory (space), and therefore, 'i-DSS' is performing better decisions during the ICV's ride.

## 5. Results and Discussion

In this section, three scenarios have been constructed to showcase the efficiency of the proposed selection process in 'i-DSS', by analyzing various aspects which include the computational effort required in various situations and the DS selections conducted. For this purpose, SimEvents (an add-on to MATLAB [42] for discrete-event simulations) is

being applied for the evaluation and validation of our 'i-DSS', mostly in terms of speed of convergence and accuracy.

*5.1. Simulation Setup and Assumptions*

As mentioned previously, three scenarios were implemented, aiming to showcase how 'i-DSS' can find the optimal DS during the ICV's ride. More in detail, the first scenario represents a "regular" case, whereas the second scenario takes into account that a specific input parameter is of high interest. The third scenario provides evidence on how 'i-DSS' and its solution process are adjusted, as fast as possible, to a situation that changes during the road trip. In all the scenarios the simulation process is applied every 0.1 s in 30 series of iterations (discrete events).

Simulation scenarios presented in the current study are quite different in providing personalized decisions for drivers with specific personal preferences and external environments with different characteristics, compared with other existing simulations in the literature [15–18,24,25]. As a result, the functionality presented herein adds value to the results extracted by the aforementioned simulations in showcasing the efficiency of 'i-DSS', mostly in terms of accuracy and speed of convergence, and adjusting clearly the actual research findings.

Moreover, the applied simulation framework is based on the assumption that each event takes place at a specific moment in time, and after a discrete sequence of events, a specific change in DS is marked. These changes are based on the input values of the parent nodes (ROC, ROT, ECO, COM, and VER). In this respect, input values for ECO, COM, and VER parameters are based on the described process in Section 3, where users (drivers or/and passengers) can voluntarily participate in evaluating the behavior of the 'i-DSS' functionality towards the in-vehicle designed set of driving styles. A grid of five (discrete) reference states is taken into consideration (see Figure 2), which expresses the level of satisfaction of each driver, i.e., "not at all satisfied" ($rs_{ij}^1 = 1$), "low satisfied" ($rs_{ij}^2 = 2$), "average satisfied" ($rs_{ij}^3 = 3$), "high satisfied" ($rs_{ij}^4 = 4$), and "completely satisfied" ($rs_{ij}^5 = 5$), where $i = \{1, 2, 3\}$ and $j = \{3, 4, 5\}$.

Furthermore, input values for ROT and ROC parameters are based on another process, which is also described in Section 3, where the ICV's in-vehicle operator system evaluates continuously (in an automated manner) the performance of the available driving styles in tackling the external driving environment. In the same way, a grid of five (discrete) reference states is taken into account (see Figure 2), which expresses the ability of the available driving styles to tackle driving environment situations, i.e., "extremely low response" ($rs_{ij}^1 = 1$), "quite low response" ($rs_{ij}^2 = 2$), "average response" ($rs_{ij}^3 = 3$), "quite high response" ($rs_{ij}^4 = 4$), and "extremely high response" ($rs_{ij}^5 = 5$), where $i = \{1, 2, 3\}$ and $j = \{1, 2\}$.

Additionally, importance plays an important role in the whole simulation process. In this respect, the driver can specify the weight values for ECO, COM, and VER parameters, before he/she starts the road journey with their ICV. On the other hand, the operator system specifies the weights for ROT and ROC parameters, which represent the external driving environment.

Regarding the limitation of the solution approach, let it be noted that the functionality's response has been tested in a small number of test scenarios, and therefore, the relative simulation results cannot be generalized. Furthermore, the overall solution analysis is based on the NB classifier, where the input features (ROC, ROT, ECO, COM, and VER) are supposed to be statistically independent.

*5.2. 1st Scenario: Normal/Regular Case*

The first scenario is related to a regular case and aims to reflect the efficiency of 'i-DSS', by identifying the best available DS during the ride. In this case, an individual driver is considered, who has already registered on 'i-DSS', by stating their personal preferences towards a predefined set of QoS features and filling in their profile data. As depicted in

Table 1, the driver provides high importance (0.3) to ECO and VER parameters, while low importance (0.15) has given to COM.

**Table 1.** 1st Scenario: Contextual parameters, collected evaluation values, and respective weight values.

| #j | Contextual Parameter | Notation | Weight Value | Input Collected Value through the Evaluation Procedures | | |
|---|---|---|---|---|---|---|
| | | | | *DS* = 1 | *DS* = 2 | *DS* = 3 |
| 1 | Road Type | ROT | 0.15 | 4 | 4 | 4 |
| 2 | Road Condition | ROC | 0.1 | 5 | 5 | 5 |
| 3 | Comfort | COM | 0.15 | 3 | 3.5 | 4 |
| 4 | Economy | ECO | 0.3 | 2.5 | 3 | 4.7 |
| 5 | Vehicle Reaction | VER | 0.3 | 3.4 | 3.7 | 4 |

The driver wishes to travel with his ICV from the departure point SP-1 to the final destination point DP-1, as shown in Figure 4. ICV is moving on a big road (well-maintained with two lines per direction), and therefore, as road condition is extremely good, the ROC parameter has a low importance (0.1), whereas ROT has higher importance (0.15), with respect to the big road with two lines per direction. Following the weight values depicted in Table 1, their sum is equal to 1. Additionally, in Table 1, mean collected values towards the five input parameters ROC, ROT, ECO, COM, and VER are demonstrated for each one of the three available styles (*DS* = 1, 2, and 3), based on the available data sets from the evaluation procedures described previously in Section 3.

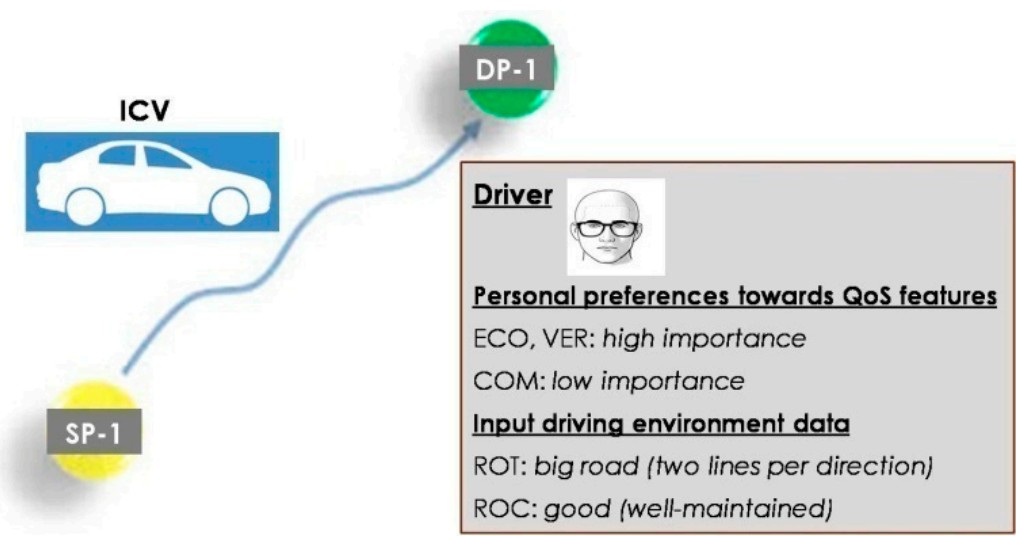

**Figure 4.** 1st Scenario: "regular" case.

Initially, as no previous knowledge is available, the probabilities $Pr[V_j = rs_{ij}^k \,|\, DS = i]$ for each input variable j are equal to 0.2. Moreover, the prior probabilities $Pr[DS = 1]$, $Pr[DS = 2]$ and $Pr[DS = 3]$ take the value 0.333 due to the fact that the same amount of information is existed for each available state of DS.

With respect to the cognitive evolution of the conditional probabilities, Figure 5a,b present the relative distributions for the parameters ECO ($j = 4$) and VER ($j = 5$) that can be obtained by $DS = 3$ (defensive driving style). The diagrams in Figure 5 show that 'i-DSS' can reach the certain states of ECO and VER parameters after a few iterations (discrete events), by converging to the mean collected in Table 1. On this basis, the conditional probability $Pr[V_5 = rs_{35}^4 \,|\, DS = 3]$ for the VER parameter becomes significant very soon, and its values are much higher than the conditional probabilities $Pr[V_5 = rs_{35}^5 \,|\, DS = 3]$ and $Pr[V_5 = rs_{35}^3 \,|\, DS = 3]$ for the "neighboring" states. Additionally, a slight diminishment is observed for $Pr[V_5 = rs_{35}^2 \,|\, DS = 3]$ from the beginning, and a severe degradation is noticed

for $Pr[V_5 = rs_{35}{}^1 | DS = 3]$. Similar considerations can be stated about the ECO parameter. Figure 5b shows that the 'i-DSS' converges fast and successful to the mean collected value (4.7), as shown in Table 1.

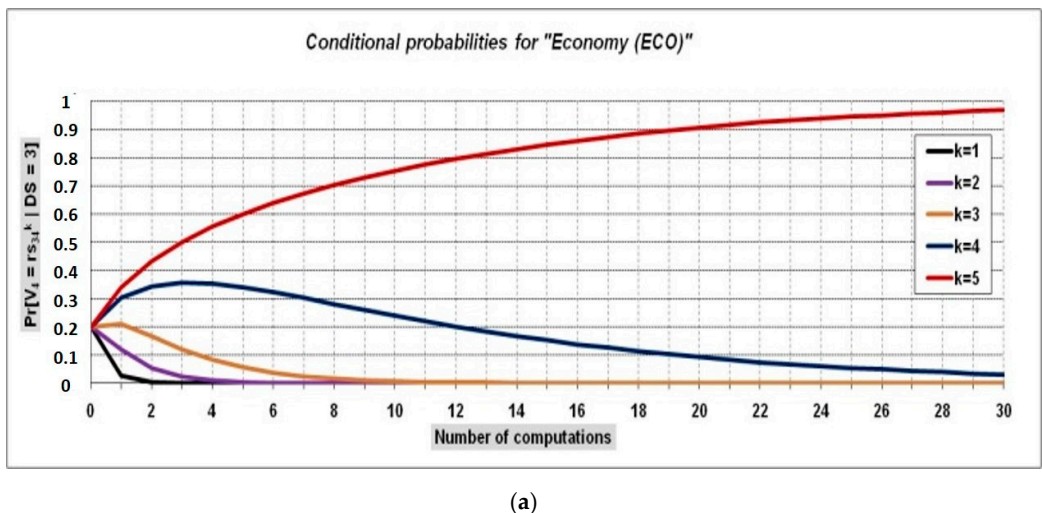

(a)

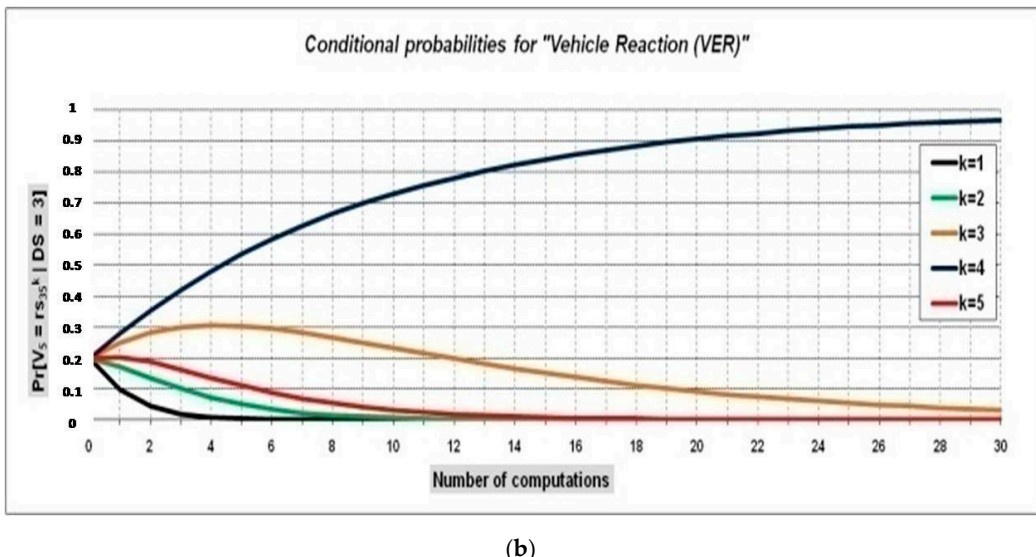

(b)

**Figure 5.** 1st Scenario: Defensive driving style (DS = 3) and conditional probabilities curves for the QoS parameters (a) ECO and (b) VER.

Due to changes in the conditional probabilities, the $f(x_i)$ values are also changing, as shown in Figure 6 for the available set of driving styles (*DS* = 1, 2, and 3). The relative curves depict that the gradual acquisition of knowledge is more "easier" for the *DS* = 3 (defensive driving style), as it becomes significant after a small amount of computations (five to six). On the other hand, there is a significant delay in the increase in the gradual acquisition of knowledge for *DS* = 1 (aggressive driving style), whereas the $f(x_i)$ curve for *DS* = 2 (normal driving style) is between the other two options of DS.

The simulation process continues with the behavior of the selection process, by taking into account the effect of the weight values on the decision-making phase. On this basis, the OF values are calculated by using (7) for all the designed driving styles. All the OF variations are shown in Figure 7. According to the depicted results, it can be stated that *DS* = 3 (defensive driving style) is the most appropriate DS, according to the input parameter and the relative policy information. A small number of iterations is required for the solution process in 'i-DSS' to take reliable decisions for the 1st Scenario towards the optimal DS.

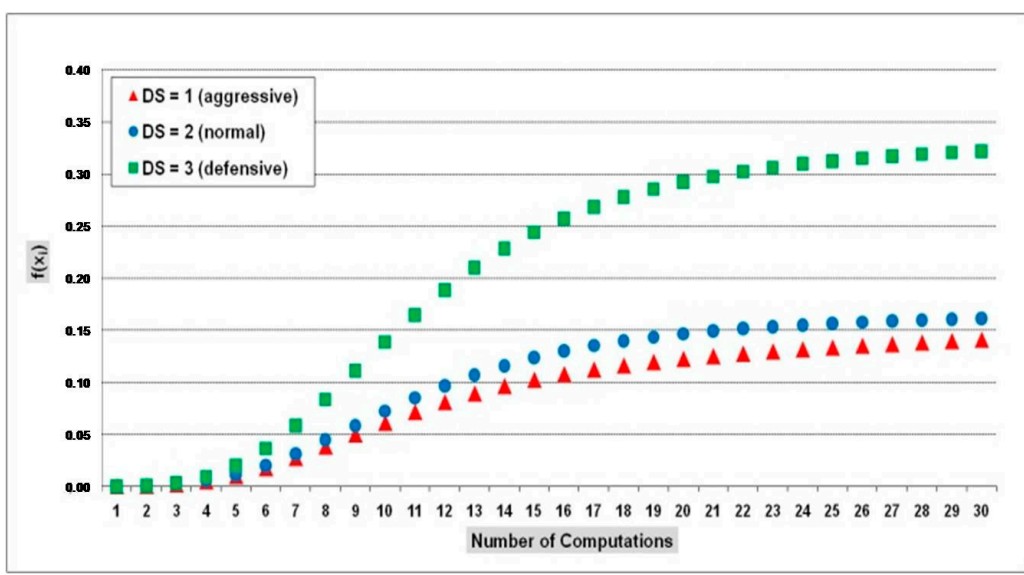

**Figure 6.** 1st Scenario: f($x_i$) curves for the three in-vehicle available driving styles.

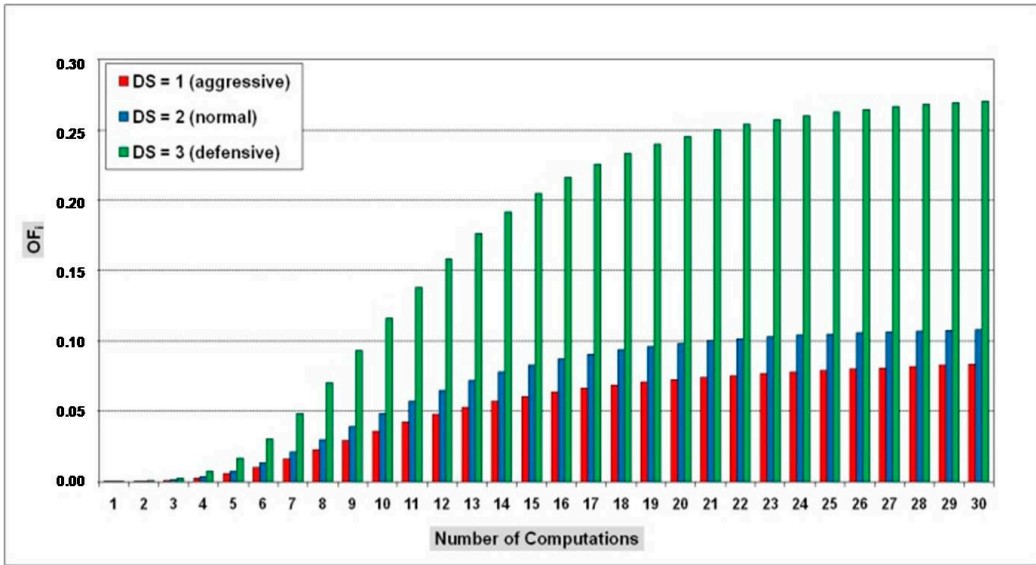

**Figure 7.** 1st Scenario: Objective Function (OF) values for the three in-vehicle available driving styles.

*5.3. 2nd Scenario: Economy Case*

This scenario aims to show the efficiency of the proposed 'i-DSS' when the driver provides a high weight value to a specific input feature (ECO, in our case). In this scenario, a candidate driver wishes to travel with their ICV from the departure point SP-2 to the final destination point DP-2 (Figure 8). The ICV is going to be moved on slippery/wet national road between SP-2 and DP-2.

The candidate driver has already stated on 'i-DSS' their profile characteristics and personal preferences (through weight values) to each one of the input QoS parameters. As shown in Table 2, ECO parameter takes a very high importance value (0.5), while COM and VER parameters take lower weight values (0.1). Additionally, due to external road condition (wet/slippery state), the ICV's central operator gives a higher importance (0.25) to the ROC parameter. Furthermore, as the ICV is moving on a national road, the ROT parameter takes a lower weight value (0.05). It should be stated that all the aforementioned weight values have a sum equal to 1. In addition, Table 2 demonstrates the mean collected

values for each one of the three DS states (DS = 1, 2 and 3) per input parameter, based on the evaluation procedures from the in-vehicle operator system and the other drivers.

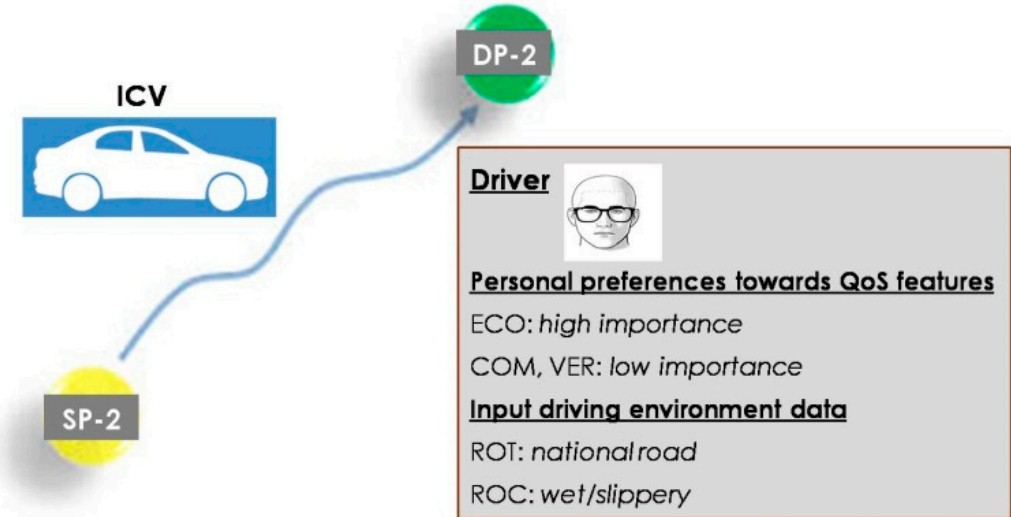

**Figure 8.** 2nd Scenario: "riding economy" case.

**Table 2.** 2nd Scenario: Contextual parameters, collected evaluation values, and respective weight values.

| #j | Contextual Parameter | Notation | Weight Value | Input Collected Value through the Evaluation Procedures | | |
| --- | --- | --- | --- | --- | --- | --- |
| | | | | DS = 1 | DS = 2 | DS = 3 |
| 1 | Road Type | ROT | 0.05 | 5 | 5 | 5 |
| 2 | Road Condition | ROC | 0.25 | 3 | 3 | 3 |
| 3 | Comfort | COM | 0.1 | 3 | 4 | 3.5 |
| 4 | Economy | ECO | 0.5 | 2.5 | 4.8 | 3 |
| 5 | Vehicle Reaction | VER | 0.1 | 3.4 | 4 | 3.7 |

The cognitive evolution of the conditional probabilities in the selection process is applied again. Indicatively, Figure 9a,b present the variations of conditional probabilities for the ECO and VER parameters with respect to the normal driving style (DS = 2). Like previously, 'i-DSS' functionality quickly converges to the collected evaluation values for the ECO and VER parameters.

Coming to the ECO parameter, the conditional probability $Pr[V_4 = rs_{24}{}^5 \mid DS = 2]$ regarding the reference state $k = 5$, becomes dominant after a few iterations. On the other hand, the probabilities for the reference values $k = 1, 2, 3$, and 4, suffer a degradation. With respect to the VER parameter, similar notes can be derived, where the reference state $k = 4$ is reached after 18 iterations. As such, 'i-DSS' provides fast and successful adaptations to the mean collected values depicted in Table 2.

The $f(xi)$ values regarding the capabilities of the driving styles according to (2), are shown in Figure 10. The curves show that the acquisition of knowledge is more difficult in $DS = 1$ and 3 (aggressive and defensive driving styles), compared to the curve of $DS = 2$ (normal driving style), due to the significant delay in their increase.

Finally, the OF variations are shown in Figure 11 for the three available driving styles (DS = 1, 2, and 3). In fact, what can be stated, is that the OF values for DS = 2 (normal driving style) becomes high very soon. This shows the ability of 'i-DSS' to efficiently aggregate the driver preferences, as the weight value for the ECO parameter is very high. Therefore, DS = 2 is the most appropriate riding economy driving style to be implemented for the second Scenario.

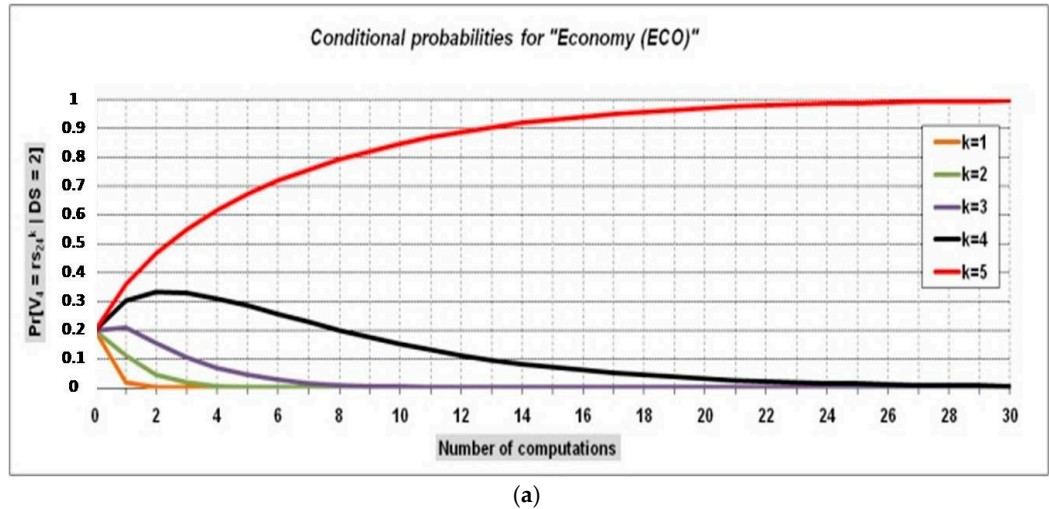

(**a**)

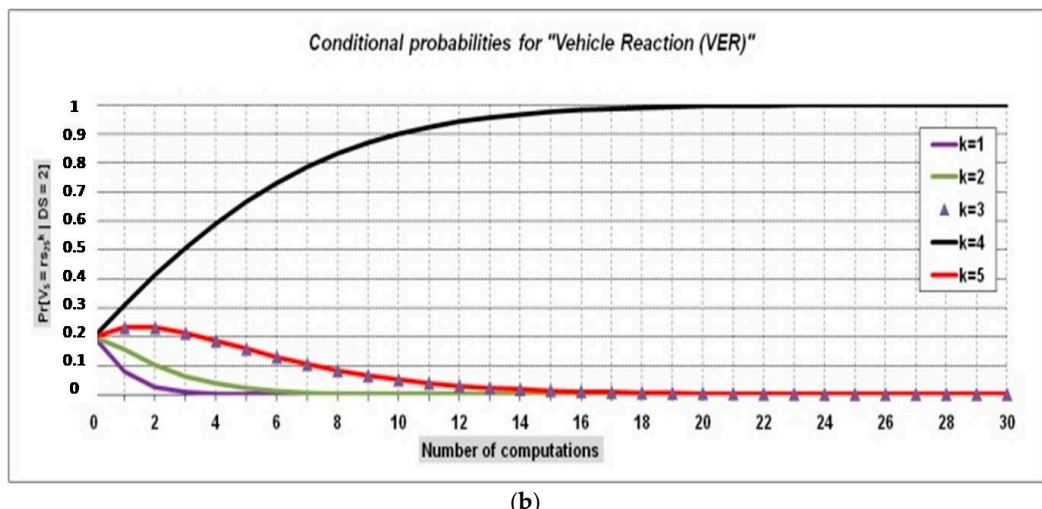

(**b**)

**Figure 9.** 2nd Scenario: Normal driving style (DS = 2) and conditional probabilities curves for the parameters (**a**) ECO and (**b**) VER.

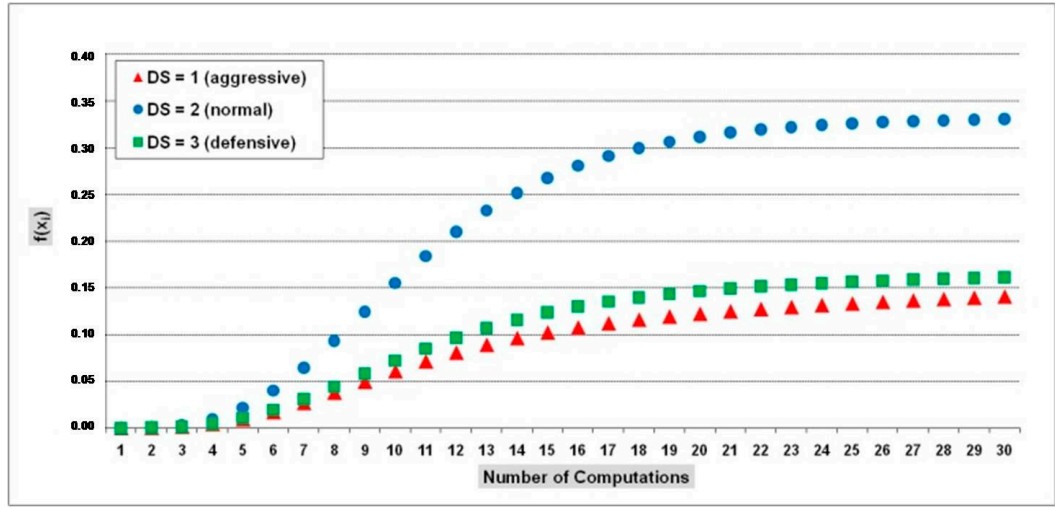

**Figure 10.** 2nd Scenario: $f(x_i)$ curves for the three in-vehicle available driving styles.

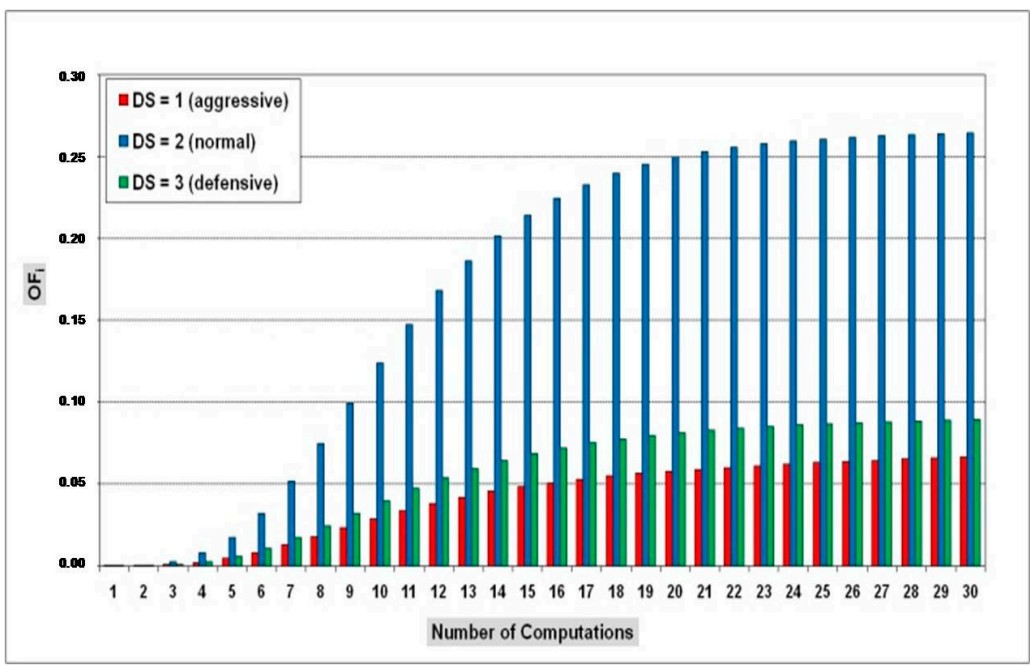

**Figure 11.** 2nd Scenario: Objective function (OF) values for the three in-vehicle available driving styles.

*5.4. 3rd Scenario: Impact of Road Condition*

In the present scenario, the scope is to show the efficiency of the 'i-DSS', in case that an external driving environment feature (ROC in our case) changes during the ride mission. In this respect, a candidate driver has a journey with their ICV, from SP-3 to DP-3, on a big urban road. As shown in Figure 12, up to point C, the ROC is good, whereas after C low-intensity rain is existed, and therefore, the ROC parameter changes to wet/slippery condition.

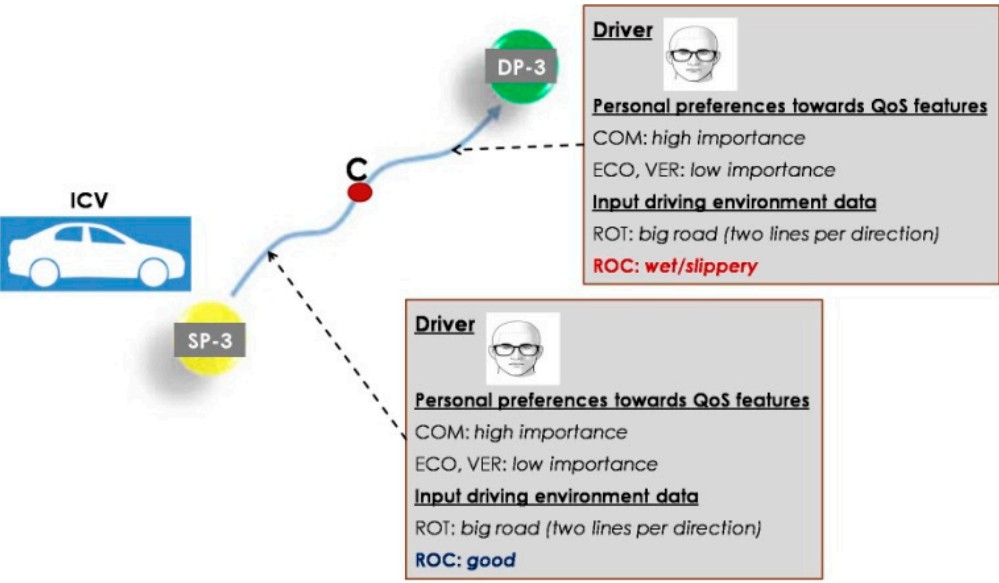

**Figure 12.** 3rd Scenario: "impact of road condition" case.

Table 3 shows the mean collected values and the respective weight values for the available set of input variables. The ICV's central operator system provides low importance (0.1) to the ROT parameter (big urban road), whereas the importance for the ROC parameter

is high (0.3), as there are changes in weather during the journey. On the other hand, the candidate driver gives high importance value (0.3) to the COM parameter and lower importance values in ECO and VER parameters.

**Table 3.** 3rd Scenario: Contextual parameters, collected evaluation values, and respective weight values.

| #j | Parameter | Notation | Weight | Input Collected Value through the Evaluation Procedures | | | | | |
|---|---|---|---|---|---|---|---|---|---|
| | | | | DS = 1 | | DS = 2 | | DS = 3 | |
| | | | | 1st | 2nd | 1st | 2nd | 1st | 2nd |
| 1 | Road Type | ROT | 0.1 | 4 | 4 | 4 | 4 | 4 | 4 |
| 2 | Road Condition | ROC | 0.3 | 5 | 2 | 5 | 3 | 5 | 4 |
| 3 | Comfort | COM | 0.3 | 3 | 3 | 3 | 3 | 4 | 4 |
| 4 | Economy | ECO | 0.15 | 4.2 | 4.2 | 3.8 | 3.8 | 2.8 | 2.8 |
| 5 | Vehicle Reaction | VER | 0.15 | 4 | 4 | 3.7 | 3.7 | 3.4 | 3.4 |

In Figure 13, the variations of conditional probabilities for the ROC parameter are presented, by taking into consideration for the $DS = 3$. Two main phases can be derived from the relative curves. First of all, in the 1st phase (iterations 1–10), the probability $Pr[V_2 = rs_{32}{}^5 \mid DS = 3]$ is the dominant one (depicted with blue line). On the other hand, in the second phase (iterations 11–20), the probability $Pr[V_2 = rs_{32}{}^5 \mid DS = 3]$ takes high values, but the probability $Pr[V_2 = rs_{32}{}^4 \mid DS = 3]$ is gradually increasing (depicted with green line). After the twentieth iteration, the probability $Pr[V_2 = rs_{32}{}^4 \mid DS = 3]$ becomes the prevalent one. Based on the above, it seems that the embedded solution process in 'i-DSS' can efficiently adapt to the new value of the ROC parameter for $DS = 3$ (defensive driving style), as a small amount of iterations is required.

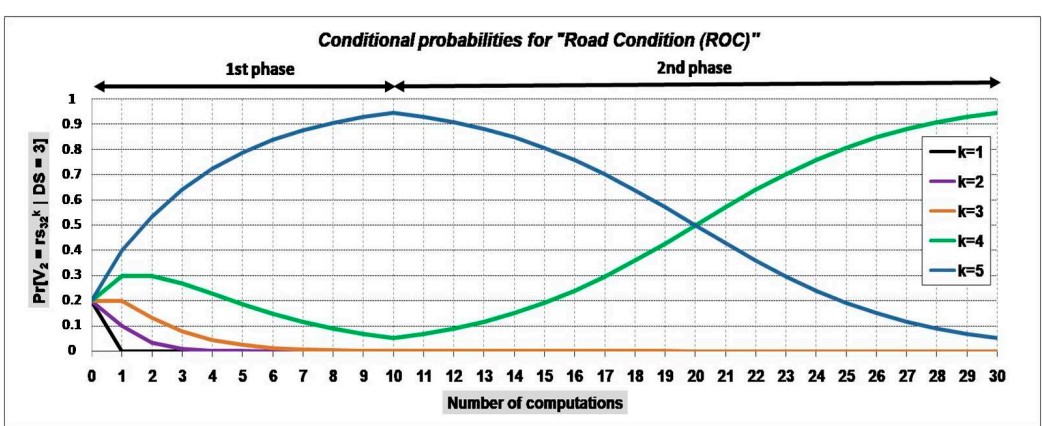

**Figure 13.** 3rd Scenario: Conditional probabilities of the ROC parameter towards the defensive driving style (DS = 3) in the two phases (1st and 2nd).

Finally, the OF values were calculated based on (2), as shown in Figure 14. The depicted variations show that DS = 3 is dominant after almost 26 iterations, whereas $DS = 1$ was dominant for the first ten iterations. Similar like previously, a small computational effort is required for the transition of DS from the aggressive state ($DS = 1$) to the defensive state ($DS = 3$). On this basis, the proposed 'i-DSS' is tested under demanding situations, as one input feature (ROC in our case) changes during the travel, and therefore, 'i-DSS' can efficiently provide the best available DS for the Scenario 3.

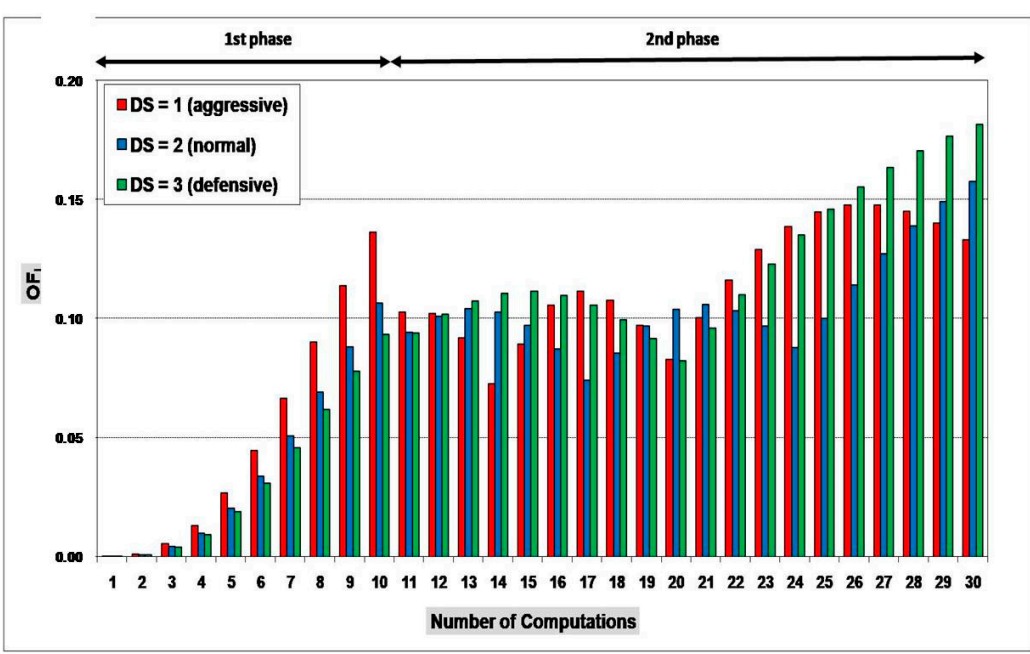

**Figure 14.** 3rd Scenario: Objective Function (OF) values for the three in-vehicle available driving styles in the two phases of the solution process.

## 6. Conclusions and Future Directions

ICVs are expected to dramatically change ground transportation and the way in which we will move in the future. In order to maximize the benefits that ICVs can bring to the mobility sector, fast, secure, and reliable cognitive systems are required. Cognitive personalized platforms seem a promising way to enhance the driving experience with ICVs, opening new gates for efficient and reliable intelligent transportation systems. Such platforms can dynamically reconfigure their operation, being able to adapt to changes in the external driving environment.

In this context, this paper has presented a driver-centric in-vehicle autonomous decision-making functionality ('i-DSS'), which enables ICVs to take real-time intelligent decisions and operate in the best available DS. Therefore, 'i-DSS' aims to enhance the efficiency of ICVs in terms of the riding safety and vehicle comfort. The solution process is based on the Bayesian networks theory and specifically the NB learning classifier. Extensive simulations for drivers with different personal preferences and external environments with changing situations are applied to show the efficiency of the proposed 'i-DSS'.

Lastly, future research plans have already been put in place, so as to expand the results of the present analysis. Firstly, decision-making analysis with well-defined, supervised ML techniques can be implemented for obtaining the appropriate DS decisions. Secondly, driving simulators or field tests can be applied to investigate the driver-functionality interactions and showcase the efficiency of 'i-DSS'. All these simulations, with regards to drivers with specified personal preferences and to external scenes with changing situations, will be of crucial importance for the training and further implementation of 'i-DSS' to ICVs. Finally, appropriate risk assessment frameworks can be applied to the solution process in order to enhance the quality of DS decisions.

**Author Contributions:** Conceptualization, I.P. and G.D.; methodology, I.P. and G.D.; software, I.P.; validation, I.P.; formal analysis, I.P.; investigation, I.P.; resources, I.P.; data curation, I.P.; writing—original draft preparation, I.P.; writing—review and editing, I.P. and G.D. All authors have read and agreed to the published version of the manuscript.

**Funding:** This research received no external funding.

**Institutional Review Board Statement:** Ethical review and approval were waived for this study based on the basis that this type of study is non-human subject research.

**Informed Consent Statement:** Informed consent was obtained from all subjects involved in the study.

**Data Availability Statement:** The data sets generated during the current study are available from authors on reasonable request.

**Conflicts of Interest:** The authors declare no conflict of interest.

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
