# Peer review of "Intelligent, In-Vehicle Autonomous Decision-Making Functionality for Driving Style Reconfigurations"

_electronics, doi:10.3390/electronics12061370_

Round 1

Reviewer 1 Report

This paper addresses the driving style adaptation for intelligent vehicles.

This aspect is important to allow ADAS-equipped and Autonomous vehicles to behave seamlessly within an environment where fully human-driven, assisted-driving, and autonomous-driving vehicles co-exist.

Moreover, the paper also considered aspects related to the social acceptance of autonomous and semi-autonomous (L2) vehicles. About this topic, there is a previous paper considering only the autonomous driving case already published in this journal, https://www.mdpi.com/2079-9292/9/3/518

Author Response

Comment 1.1: This paper addresses the driving style adaptation for intelligent vehicles. This aspect is important to allow ADAS-equipped and Autonomous vehicles to behave seamlessly within an environment where fully human-driven, assisted-driving, and autonomous-driving vehicles co-exist. Moreover, the paper also considered aspects related to the social acceptance of autonomous and semi-autonomous (L2) vehicles. About this topic, there is a previous paper considering only the autonomous driving case already published in this journal, https://www.mdpi.com/2079-9292/9/3/51.

Response 1.1: We would like to thank the reviewer for this very interesting suggestion. In fact, we have added a reference to this paper (now [35]) and we explain the fact that our paper is based also on this work, building upon it.

Reviewer 2 Report

Comments to Authors:

·         The paper needs much improvement in English.

·         The novelty of your approach is not evident in your introduction; show the novelty clearly.

·         Clearly state the contributions in a list in the introduction section. Because your contributions list only states what you have done, not what your approach achieves.

·         Add a section named “Literal Review and Background Work” and move the contents of the literature review from the introduction to this section. There are the following ml approaches related to your topic:

o   Lv, Chen, et al. "Driving-style-based codesign optimization of an automated electric vehicle: A cyber-physical system approach." IEEE Transactions on Industrial Electronics 66.4 (2018): 2965-2975.

o   Liu, Guangqun, et al. "Deep learning-based channel prediction for edge computing networks toward intelligent connected vehicles." IEEE Access 7 (2019): 114487-114495.

o   Fang, Weidong, et al. "BTDS: Bayesian‐based trust decision scheme for intelligent connected vehicles in VANETs." Transactions on Emerging Telecommunications Technologies 31.12 (2020): e3879.

o   Li, Chunhai, et al. "Distributed perception and model inference with intelligent connected vehicles in smart cities." Ad Hoc Networks 103 (2020): 102152.

o   Xie, Bowen, et al. "MOB-FL: Mobility-Aware Federated Learning for Intelligent Connected Vehicles." arXiv preprint arXiv:2212.03519 (2022).

·         Remove the “Motivation and High-Level Description” section and move some of the content to “Literal Review and Background Work.”

·         Remove the “Overall Solution Approach” section and move some of the content to “Literal Review and Background Work.” and “Methodology” section.

·         In the “Literal Review and Background Work.” Add the background information about Bayesian networking, notably the Nave-Bayes learning classifier.

·         Add a comparison table in the “Literal Review and Background Work.” with the related approaches in terms of approach used, features, etc..

·         Add a section called “Problem Description and Formulation” where you describe and formulate the problem. Move the problem formulation from the model section to this section. The provided problem formulation is very poor… enrich it with the transition formulas.

·         Add a section called “Problem Description” where you describe the problem.

·         Add a “Methodology” section explaining your approach and move the rest of the content from “Motivation and High-Level Description” to this section.

·         Add a subsection for your assumptions under the “Results and Discussion” section.

·         At “Motivation and High-Level Description”, you have Figure -1, the i-DSS architecture. Please explain each component in the figure and provide a Sequence Diagram (UML) that shows the interaction among each component. Moreover, you should provide a class diagram to show the data representation.

·         You should add in the new “Methodology” section a full description of the dataset you used and a complete description of the parameters used as sources resulting in being featured in the ML model. How did you gather the data for the dataset creation?

·          Moreover, in the methodology under the Dataset creation, you should include the feature selection process. Which features are at the end to be used in the model?  You need to execute a feature extraction process to identify the most appropriate features that will result in your approach training.

·         Formula 4 is with Chinese letters; please write it with English.

·         Explain in depth why your approach achieves more accuracy than other approaches.

·         Add comparison figures with other approaches found in Literature Review and add a table to show your approach's novelty in the “Results and Discussion” section.

·         Explain Figures 4-14 in more depth, explaining why the investigated approach concluded with the show results. Also, show how parameters/features are affected.

·         In the conclusion section, add your future work.

Author Response

Comment 22.1: The paper needs much improvement in English.

Response 2.1: We have revised the whole paper, reworking several parts of it and improved the English accordingly. For this, we have also got the help of a native English speaker.

Comment 2.2: The novelty of your approach is not evident in your introduction; show the novelty clearly.

Response 2.2: In the revised version of the paper we have reworked the introduction, clearly stating that we build upon previous research attempts and describe a novel functionality for the dynamic reconfiguration of driving styles, incorporating knowledge and experience.

Comment 2.3: Clearly state the contributions in a list in the introduction section. Because your contributions list only states what you have done, not what your approach achieves.

Response 2.3: We have reworked the section "1. Introduction" and now we clearly state that: “In order to respond to the aforementioned challenges, the contribution of this paper lies in that it presents a novel in-vehicle decision making functionality, able to proactively, efficiently and securely decide on the most appropriate DS, in dynamically changing environments, considering all the driver’s personal preferences, as well as contextual parameters from the vehicle’s environment. Let it also be noted the proposed functionality acts in a fully autonomous (self-adaptive) manner, requiring no driver intervention.”

Comment 2.4: Add a section named “Literal Review and Background Work” and move the contents of the literature review from the introduction to this section.

Response 2.4: We would like to thank the reviewer for this comment. In the revised version of the manuscript, we have added this section.

Comment 2.5: Add a • There are the following ml approaches related to your topic:

    • Lv, Chen, et al. "Driving-style-based codesign optimization of an automated electric vehicle: A cyber-physical system approach." IEEE Transactions on Industrial Electronics 66.4 (2018): 2965-2975.
    • Liu, Guangqun, et al. "Deep learning-based channel prediction for edge computing networks toward intelligent connected vehicles." IEEE Access 7 (2019): 114487-114495.
    • Fang, Weidong, et al. "BTDS: Bayesian‐based trust decision scheme for intelligent connected vehicles in VANETs." Transactions on Emerging Telecommunications Technologies 31.12 (2020): e3879.
    • Li, Chunhai, et al. "Distributed perception and model inference with intelligent connected vehicles in smart cities." Ad Hoc Networks 103 (2020): 102152.
    • Xie, Bowen, et al. "MOB-FL: Mobility-Aware Federated Learning for Intelligent Connected Vehicles." arXiv preprint arXiv:2212.03519 (2022).

 Response 2.5: We have referred to the aforementioned papers in the revised version of the paper, explaining how we took into consideration these papers in building and proposing our functionality.

Comment 2.6: Remove the “Motivation and High-Level Description” section and move some of the content to “Literal Review and Background Work”.

Response 2.6: We would like to thank the reviewer for this suggestion. We followed it exactly in the revised version of the manuscript.

Comment 2.7: Remove the “Overall Solution Approach” section and move some of the content to “Literal Review and Background Work.” and “Methodology” section.

Response 2.7: We would like to thank the reviewer for this suggestion. We followed it exactly in the revised version of the manuscript.

Comment 2.8: In the “Literal Review and Background Work.” Add the background information about Bayesian networking, notably the Nave-Bayes learning classifier.

Response 2.8: We would like to thank the reviewer for this suggestion. We followed it exactly in the revised version of the manuscript.

Comment 2.9: Add a comparison table in the “Literal Review and Background Work.” with the related approaches in terms of approach used, features, etc.

Response 2.9: We would like to thank the reviewer for this suggestion. All the related approaches in terms of approach used, features, etc. have been identified clearly in the main text within the section “2. Literature Review and Background Work”.

Comment 2.10: Add a section called “Problem Description and Formulation” where you describe and formulate the problem. Move the problem formulation from the model section to this section. The provided problem formulation is very poor… enrich it with the transition formulas.

Response 2.10: We would like to thank the reviewer for this suggestion. We followed it exactly in the revised version of the manuscript.

Comment 2.11: Add a section called “Problem Description” where you describe the problem.

Response 2.11: We would like to thank the reviewer for this suggestion. We followed it exactly in the revised version of the manuscript.

Comment 2.12: Add a “Methodology” section explaining your approach and move the rest of the content from “Motivation and High-Level Description” to this section.

Response 2.12: We would like to thank the reviewer for this suggestion. We followed it exactly in the revised version of the manuscript.

Comment 2.13: Add a subsection for your assumptions under the “Results and Discussion” section.

Response 2.13: We would like to thank the reviewer for this suggestion. We followed it exactly in the revised version of the manuscript.

Comment 2.14: At “Motivation and High-Level Description”, you have Figure -1, the i-DSS Please explain each component in the figure and provide a Sequence Diagram (UML) that shows the interaction among each component. Moreover, you should provide a class diagram to show the data representation.

Response 2.14: We would like to thank the reviewer for this suggestion. In the revised version of the manuscript, each component in Figure 1 are explained in detail in the subsection 3.1. Figure1 shows the basic components of the overall  “i-DSS” architecture and the interactions among each component, whereas data related to each input and output component are presented in the subsection 4.1.

Comment 2.15: You should add in the new “Methodology” section a full description of the dataset you used and a complete description of the parameters used as sources resulting in being featured in the ML model. How did you gather the data for the dataset creation?

Response 2.15: We would like to thank the reviewer for this suggestion. In the revised version of the manuscript, dataset and parameters complete description we used are provided in section "5. Results and Discussion", where three scenarios have been constructed to showcase the efficiency of the proposed methodology. The present paper focuses on the evaluation and validation process of our proposed ‘i-DSS’, which includes computational effort required in various situations and the DS selections conducted. Dataset creation and data gathering have already been putted in place in our research efforts, being able to expand the evaluation and validation of the proposed methodology.

Comment 2.16: Moreover, in the methodology under the Dataset creation, you should include the. Which features are at the end to be used in the model? You need to execute a feature extraction process to identify the most appropriate features that will result in your approach training

Response 2.16: We would like to thank the reviewer for this suggestion. In the revised version of the manuscript, in the section "1. Introduction", the authors clearly state that the present paper aims to introduce and develop an autonomous (self-adaptive) management functionality where cognitive management techniques are incorporated to aggregate extensive data sources in real-time (driving surrounding context, driver’s preferences, operational requirements) and interprets them to assess whether a specific DS is appropriate in each case, using previous knowledge and experience. Feature selection process and identification of the most appropriate features that will result in our approach training have already been putted in place in our research efforts, being able to expand the evaluation and validation of the proposed methodology.

Comment 2.17: Explain in depth why your approach achieves more accuracy than other approaches.

Response 2.17: We would like to thank the reviewer for this. In the revised version of the manuscript why our approach achieves high accuracy: “Every time a contextual situation is encountered, the system compares it with information available in the database. In the case of a match above a certain percentage (e.g. 95%), the system decides on re-applying a previously successful decision, without running again the time-consuming algorithmic process described above. In a negative case, the algorithm needs to run whatsoever”.

Comment 2.18: Formula 4 is with Chinese letters; please write it with

Response 2.18: We would like to thank the reviewer for this. In the revised version of the manuscript, formula 4 has been rewritten only with English letters.

Comment 2.19: Add comparison figures with other approaches found in Literature Review and add a table to show your approach's novelty in the “Results and Discussion” section.

Response 2.19: We would like to thank the reviewer for this. Comparison figures and tables with other approaches will be part of our future research plans. This process is quite difficult due to the fact that different aspects like validation framework, feature selection process, parameters in the training, ML method, etc should be taken into consideration for this analysis. As mentioned previously, the aim of the present paper aims to present a novel in-vehicle decision making functionality, able to proactively, efficiently and securely decide on the most appropriate DS, in dynamically changing environments, considering all the driver’s personal preferences, as well as contextual parameters from the vehicle’s environment.

Comment 2.20: Explain Figures 4-14 in more depth, explaining why the investigated approach concluded with the show results. Also, show how parameters/features are affected.

Response 2.20: We would like to thank the reviewer for this. We followed it exactly in the revised version of the manuscript.

Comment 2.21: Explain In the conclusion section, add your future work.

Response 2.21: We would like to thank the reviewer for this. In the revised version of the paper, section "6. Conclusion and Future Directions" been reworked in order to present and explain more in detail the limitations and the future research areas of the present study.

Reviewer 3 Report

The paper is interesting, but there are unclear points.

In the literature review, Studies on driving styles are missing (see: DOI: 10.1177/03611981211034718; Doi: 10.1016/j.trpro.2017.12.024; DOI: 10.1016/j.aap.2022.106763).

Authors should introduce a data paragraph, describing which data are analysed, eg Brijs et al (DOI: 10.1016/j.aap.2022.106763) use lateral position and Heading to describe driving behaviors.

In the data, the authors must also introduce the number of driving analysed.

Author Response

Comment 3.1: The paper is interesting, but there are unclear points. In the literature review, Studies on driving styles are missing (see: DOI: 10.1177/03611981211034718; Doi: 10.1016/j.trpro.2017.12.024; DOI: 10.1016/j.aap.2022.106763).

Response 3.1: We have referred to the aforementioned papers in the revised version of the paper, explaining how we took into consideration these papers in building and proposing our functionality.

Comment 3.2: Authors should introduce a data paragraph, describing which data are analysed, eg Brijs et al (DOI: 10.1016/j.aap.2022.106763) use lateral position and Heading to describe driving behaviors. In the data, the authors must also introduce the number of driving analysed.

Response 3.2: We would like to thank the reviewer for this suggestion. We followed it exactly in the subsection 5.1 in the revised version of the manuscript.

Round 2

Reviewer 3 Report

The authors responded to all reviews

Author Response

A thorough review in the complete manuscript has been carried out, in which grammatical typos were corrected and existing grammatical and punctuation mistakes were removed.